# Conductive Bioimprint Using Soft Lithography Technique Based on PEDOT:PSS for Biosensing

**DOI:** 10.3390/bioengineering8120204

**Published:** 2021-12-09

**Authors:** Nor Azila Abd. Wahid, Azadeh Hashemi, John J. Evans, Maan M. Alkaisi

**Affiliations:** 1Department of Electrical and Computer Engineering, The MacDiarmid Institute for Advanced Materials and Nanotechnology, University of Canterbury, Christchurch 8140, New Zealand; ain3052@gmail.com (N.A.A.W.); Azy.hashemi@canterbury.ac.nz (A.H.); 2Christchurch School of Medicine and Health Sciences, University of Otago, Christchurch 8041, New Zealand; John.evans@otago.ac.nz

**Keywords:** conductive bioimprint, soft lithography, PEDOT:PSS, conductive hydrogel, electrical conductivity

## Abstract

Culture platform surface topography plays an important role in the regulation of biological cell behaviour. Understanding the mechanisms behind the roles of surface topography in cell response are central to many developments in a Lab on a Chip, medical implants and biosensors. In this work, we report on a novel development of a biocompatible conductive hydrogel (CH) made of poly (3,4-ethylenedioxythiophene):polystyrene sulfonate (PEDOT:PSS) and gelatin with bioimprinted surface features. The bioimprinted CH offers high conductivity, biocompatibility and high replication fidelity suitable for cell culture applications. The bioimprinted conductive hydrogel is developed to investigate biological cells’ response to their morphological footprint and study their growth, adhesion, cell–cell interactions and proliferation as a function of conductivity. Moreover, optimization of the conductive hydrogel mixture plays an important role in achieving high imprinting resolution and conductivity. The reason behind choosing a conducive hydrogel with high resolution surface bioimprints is to improve cell monitoring while mimicking cells’ natural physical environment. Bioimprints which are a 3D replication of cellular morphology have previously been shown to promote cell attachment, proliferation, differentiation and even cell response to drugs. The conductive substrate, on the other hand, enables cell impedance to be measured and monitored, which is indicative of cell viability and spread. Two dimensional profiles of the cross section of a single cell taken via Atomic Force Microscopy (AFM) from the fixed cell on glass, and its replicas on polydimethylsiloxane (PDMS) and conductive hydrogel (CH) show unprecedented replication of cellular features with an average replication fidelity of more than 90%. Furthermore, crosslinking CH films demonstrated a significant increase in electrical conductivity from 10^−6^ S/cm to 1 S/cm. Conductive bioimprints can provide a suitable platform for biosensing applications and potentially for monitoring implant-tissue reactions in medical devices.

## 1. Introduction

Over the past two decades, the application of micro fabrication in the field of biology has stimulated interest and experiments on cell growth on topographically modified surfaces. The term “bioimprint” was first postulated in 2006, in which a soft lithography technique was developed to capture 3D cellular morphology into a hard polymer for the purposes of improved cell imaging and formation of cell culture platforms [1,2]. Numerous studies have reported that culture platforms with topographical features influence cell orientation, migration, morphology, proliferation and differentiation. Various surface topographically defined patterns such as PDMS [3,4], glass [4], Permanox [4], methacrylate [3,5,6,7], polystyrene [3,4,8,9], poly (ethylene glycol) terephthalate-poly (butylene terephthalate), (PEGT-PBT) [10] and non-polymer materials such as Casein [11,12] have previously been reported. These studies have been used to understand cell–surface interactions, growth, adhesion, spreading, morphology, proliferation and differentiation of biological cells, as well as cell respond to anticancer drugs [8], as a response to surface topography and substrate material [2,3,4]. As an example, they have shown that cells, which would normally grow randomly on a flat surface in vitro, would align along the surface patterns if grown on parallel lines [2,3,4,9,13,14,15]. However, all of these studies were mostly focused on improved understanding of cells’ behaviour under different physical characteristics with respect to the chemical and mechanical properties. The electrical properties of the substrate and how it can be utilised for monitoring cells’ response is something which has not been looked into.

Motivated by the possibility of developing conductive biomaterials for implantable devices, tissue engineering and for electrode coating, a variety of conductive polymer hydrogels have previously been explored based on polypyrrole (PPy) [16,17,18], PEDOT [19,20,21,22,23,24] and polyaniline (PANI) [25,26]. However, the performance of any CH is dependent on a number of factors, such as the conductivity, degree of crosslinking and swelling behaviour of the hydrogel [27,28]. In the field of tissue engineering, 3D scaffolds have been developed rapidly in combination with biorecognition and analyte to explore their potential in repairing and regenerating diseased tissue [29,30]. Other studies on medical implants, such as a structural bone replacement including electrically stimulating the neural probes, have led to the development of a wide range of prosthetic and medical implant devices [31].

Conductive polymers such as polyacetylene (PA), polypyrrole (PPy), polythiophene (PT) and polyaniline (PANI) have important potential in the development of electroactive biomaterials for biosensing and bioengineering applications [32,33,34,35,36,37,38]. The development of CP hydrogel was first reported by Gilmore et al. in 1997 [39]. A polythiophene derivative, PEDOT:PSS has been widely used by scientists in the last decade for applications in supercapacitors [40], transistors [41], solar cells [42,43], batteries and bio-electronics [43,44,45], owing to its good conductivity, high stability at room temperature, flexibility as an aqueous solution and the fact that it is relatively inexpensive [32,37,38,40,41,42,43,44,46]. PEDOT:PSS-based CP is becoming an attractive material for interfacing with biological systems; it has good forming properties and is chemically stable [47,48]. The material is also well characterised with biocompatible properties in vitro/vivo, which opens up a range of opportunities for soft tissue augmentation and regeneration [20,21,22]. Hence, there is increased interest in several research programmes mainly in regulating cell behaviour through electrical stimulation and in helping to trigger cell desorption [44,49].

However, PEDOT:PSS is a water-based conductive polymer and has limited applications in imprinting processes or cell culture platforms because of its fragile and poor mechanical properties and is not suitable for tissue engineering by itself [23]. In this work, CP was mixed with colourless gelatin and glycerol to produce conductive hydrogel as a suitable cell culture platform for sensing applications. Gelatin was chosen because of its hydrophilic, biocompatible and biodegradable properties [50]. Gelatin has also shown high resolution fidelity when used for surface feature replication [51,52]. The use of glycerol was intended to enhance the electrical conductivity, decrease the water solubility as well as act as a plasticizing agent in the CH film. In addition, the tuneable properties of the developed CH polymer enable the formation of 3D patterning with a number of printing techniques. Various studies have reported that the CHs are attractive as scaffolding materials for regenerative medicine applications due to their ability to mimic physical properties of various tissues [21,46,50,53,54,55].

In this paper, we report for the first time the development of “conductive bioimprint” as a technique for producing cell-like features in CP using soft lithography technique. Conductive bioimprint is a novel approach that replicates cellular morphology with a nanometre scale resolution into a CP PEDOT:PSS. The cells’ replicas are suitable for cell culture platforms, enabling studying and monitoring cell’s behaviour as a function of their conductivity. The conductive substrates with cell-like surface topography offers a platform with good conductivity, high resolution replication fidelity, biocompatibility, stretchability and biodegradability in a substrate suitable for tissue culture engineering and implantable biosensor devices. The developed conductive bioimprints have potential for monitoring medical implants/body interface in real time and gauging their progress by observing cell conductivity. This could be achieved by monitoring cell conductivity of medical implants coated with our developed conductive hydrogel.

## 2. Materials and Methods

The following materials and chemicals were used in this work:

PEDOT:PSS with 1.3 wt.% dispersion in deionized water (483095-250G, Sigma Aldrich, St. Louis, MO, USA), Type A gelatin from porcine skin (G2500, Sigma Aldrich), microbial transglutaminase (mTg, supplied by Ajinomoto, Japan Co. Inc., Tokyo, Japan) 0.1% *w*/*w* hydroxypropylmethylcellulose (HPMC, H8384, Sigma Aldrich), 0.05% trypsin-EDTA (15400-054, Gibco, Waltham, MA, USA), 10% sodium dodecyl sulphate (SDS, L3771, Sigma Aldrich), phosphate-buffered saline (PBS) tablets (003002, Life Technologies, Carlsbad, CA, USA), Dulbecco’s modified Eagle’s medium (DMEM, 11995-056, Gibco via Invitrogen), fetal bovine serum (FBS, 30044333, Gibco Thermo Fisher Scientific), penicillin-streptomycin 100× (P/S, 15140122, Gibco via Invitrogen), Fungizone (Amphotericin, 15290018, Gibco via Invitrogen), Polydimethylsiloxane, (PDMS, Sylgard 184, Dow Corning Corp., Midland, TX, USA), ethanol and glycerol (analytical grade, from LabServ), Hydrochloric Acid (HCl, H1758, Sigma Aldrich), hydroxypropylmethylcellulose (HPMC, H8384, Sigma Aldrich), AZ 1518 photoresists (Micro Chemicals GmbH, Ulm, Germany), AZ 326 MIF developer (Micro Chemicals, GmbH), Hexamethyldisilazane (HMDS, 440191-1L, Sigma-Aldrich), microscope glass slide 25 mm × 75 mm (Corning), acetone, methanol and isopropanol (Sigma Aldrich).

The two main materials in this work are PEDOT:PSS and gelatin. PEDOT:PSS was chosen due to its transparency, high conductivity and water-solubility, while gelatin was chosen because it has the same properties except for conductivity to be added to PEDOT:PSS. The goal was to achieve a biocompatible CH substrate with high resolution surface bioimprints which is suitable for cell conductivity monitoring.

### 2.1. Cell Culturing and Grid Pattern Fabrication on Glass Substrates

In order to facilitate a direct comparison between the original cells and replica features, early passages of the C2C12 cells (mouse myoblast cell-line) were cultured on grid-patterned microscope slide glass. C2C12 cells were obtained and cultured according to the Christchurch School of Medicine and Health Sciences, University of Otago, New Zealand cell culture protocol guidelines. The fabrication process of the grid-patterned glass substrate followed the conventional photolithography method reported by our group [11]. The fabrication process was performed in the Nanofabrication laboratory at University of Canterbury. In the first step, glass substrate was cleaned using acetone, methanol and isopropanol for 3 min in an ultrasonic bath. It was then treated with oxygen plasma for 10 min at 100 W. The substrate was then spin coated with AZ 1518 positive photoresist at 1000 rpm for 60 s. This step was followed by a soft-baking process for 10 min at 95 °C on a hotplate. The grid patterns were defined under UV light exposure for 15 s using Karl Suss MA-6 mask aligner. The exposed photoresist was then developed in the MIF 326 developer and washed with deionized water. The developed substrate was then hard-baked on a hotplate at 120 °C for 10 min. In order to transfer the grid patterns into the glass substrate, the substrate was etched with buffered hydrofluoric acid for 2 min using the photoresist as masking material. Finally, the photoresist was removed using acetone and the slide was sterilized under UV light for at least 30 min before being used as a cell-culture substrate.

In this work, the myoblast cell line was selected owing to robust baseline viability and it provided high proliferation rates [56]. The cell line also has the capability to differentiate into myocyte or myofibrils as a response to its environment and the surface it grows on. Due to fusion and delineation of C2C12 cells, they are widely used for studying their growth, organization, and differentiation, in response to surface patterns [57].

C2C12 cells were plated on the glass slide with a seeding density of 15,000 cells/cm^2^ and grown in Dulbecco’s modified Eagle’s medium (DMEM) supplemented with 10% FBS, 1% Fungizone and 1% P/S. The cultures were incubated at 37 °C in 5% CO_2_ until they reached 80% confluency. They were then fixed with 2.5% glutaraldehyde in PBS for 2 h. A dehydration cycle of immersing cells in 75%, 85%, 95% and 100% ethanol, each for 5 min, followed the fixation and cells were finally immersed in drying agent of HMDS for 5 min before leaving them to air-dry in a fume hood in order to carry on the bioimprinting process.

### 2.2. The PDMS Bioimprint Master Mould

PDMS is widely used in the fabrication and prototyping due to having a high-resolution fidelity, being optically clear, hydrophobic, hyper elastic and biocompatible [58]. The positive(protruding) and negative(depression) bioimprint master moulds with the required patterns were made from PDMS and prepared according to a method reported by our group [10] and is illustrated in Figure 1. The PDMS elastomer and curing agent were mixed with a ratio of 10:1 *w*/*w* and stirred thoroughly by hand for 30 s. The PDMS mixtures were then placed in a desiccator for 30 min to remove any air bubbles before being poured on the fixed cells, as illustrated in Figure 1a. The curing process is set for 12 h and is accomplished at 37 °C to prevent any thermal damage to the cells, as shown in Figure 1b. As in Figure 1c, the imprinted polymer was then peeled off, and baked for another 2 h at 80 °C for further crosslinking. This produced a negative PDMS mould, which had the cellular morphology replicated on the surface as indentations. This mould was washed in 10% *w*/*w* SDS in 0.01 M HCl and 0.05% trypsin-EDTA in PBS for 3 min to remove any biological debris on the polymer surface. Final rinsing was carried out with deionized water for 3 min to remove any unwanted acidic monomer on the mould.

One of the main elements for a successful bioimprinting is dependent on the surface energy between mould and the substrate. The PDMS mould surface energy must be lower than the glass substrates surface energy, so that the polymer film will have strong adhesion to the substrate and can easily be peeled off from the mould [59]. Here, the surface of negative PDMS mould was treated with an anti-sticking layer by being immersed in 0.1% *w*/*w* HPMC solution in PBS for 10 min, as shown in Figure 1d, to avoid the negative mould from sticking to the second PDMS mould. The negative PDMS mould was then removed, washed with DI-water and dried with nitrogen. Liquid PDMS was mixed with curing agent with 10:1 *w*/*w* ratio and degassed again and was then poured onto negative PDMS mould and cured at 37 °C for 12 h (Figure 1e). The second replication from a negative mould results in a positive bioimprint mould, where cellular morphology appears the same as the fixed cells. Positive PDMS mould was then carefully peeled off from the negative bioimprint which was followed by a further bake at 80 °C for 2 h to complete the polymerization process as shown in Figure 1f.

### 2.3. The Fabrication of Plasticized PEDOT:PSS Bioimprint

To study the effect of adding glycerol to the PEDOT:PSS mixture, 6 different concentrations of glycerol (0%, 0.5%, 1%, 2%, 3% and 4% *w*/*w*) were used to synthesize plasticized PEDOT:PSS by following the method developed by Meier et al. [43]. Changes in electrical conductivity, film thickness, surface morphology and performance of the bioimprint replication process were then studied as a result of different concentrations of glycerol. The results of this study will be presented in the following sections.

In this synthesis, the water-based PEDOT:PSS solution was agitated in an ultrasonic bath for 15 min at room temperature and filtered using 5 µm PTFE filters in a further purification process. Then, different concentrations of glycerol as a plasticizer agent were added to the purified PEDOT:PSS and stirred thoroughly using a magnetic flea until a homogeneous solution was formed. The homogeneous mixture of PEDOT:PSS was then spun coated on an oxygen plasma treated microscope glass substrate at 1000 rpm for 60 s with a Headway PWM32-PS-R790 spinner machine. The coated glass substrate was then baked at 80 °C on hotplate until fully dry (the bake time was dependant on the glycerol content). Then, a PDMS mould with negative bioimprints was treated with oxygen plasma at 100 W for 30 s to establish a more hydrophilic surface. The mould was then put in contact with the plasticized coated substrate and a constant pressure of 40 kPa was applied for 48 h at 37 °C on a hot plate. This method is called plasticised-assisted soft embossing (PASE), which is a soft embossing technique that uses polymeric stamps instead of hard surfaces such as aluminium, glass or silicon [60,61,62]. This low-cost and simple method opens up a wide range of polymeric material applications including reducing the problem of breaking the mould and increasing mould reusability [59].

### 2.4. Fabrication of Conductive Hydrogel Bioimprint

CH was synthesised by mixing plasticized PEDOT:PSS with gelatin solution. A concentration of 12.5 wt.% of type A gelatin from porcine skin powder was added to 1× PBS solution (pH 7.4) and heated at 60 °C for 15 min or so until the gelatin powder was disolved in the solvent. The gelatin solution was mixed with six different equal concentrations of glycerol and PEDOT:PSS: 0%, 1%, 2%, 3%, 4% and 6% *w*/*w* (e.g., 1% is 1% of glycerol, 1% of PEDOT:PSS and 98% of gelatin solution) to investigate the changes in film transparency and conductivity. The conductive mixtures were continuously stirred at 70 °C until they became homogeneous. CH mixtures were chemically crosslinked with microbial transglutaminase (mTg) with an activity of 1000 U/g crosslinker to gain a higher mechanical strength and degradation resistance. To crosslink films, 0.5 g of mTg was dissolved with 20 mL of 1× PBS solution to achieve a 2.5% *w*/*v* mTg solution. This solution was then mixed with CH mixture. Since the concentration of gelatin in the CH mixture was 12.5 g, the final concentration of mTg in the CH mixture was 40 U of mTg per gram of gelatin.

Positive and negative replicas of C2C12 cells were made into CH films to study the suitability of the material in terms of bioimprinting process and the resolution of features. Figure 2 shows the fabrication process of positive and negative conductive bioimprints off PDMS. Three concentrations of glycerol and PEDOT:PSS in crosslinked CH mixtures were studied: 2%, 3% and 4%. Prior to the bioimprinting process, the PDMS moulds were oxygen plasma treated at 100 Watt for 30 s to establish a more hydrophilic surface before the imprinting process. The crosslinked mixtures were immediately poured directly onto the positive and negative PDMS bioimprint mould (light blue) and dried in the incubator at 37 °C overnight (Figure 2a). The conductive bioimprint material was then kept at 4 °C for 48 h to complete the polymerization. The positive and negative conductive bioimprinted films (dark blue) were then carefully peeled off to avoid tearing effects as shown in Figure 2b. As shown in Figure 2c, a second replication of negative mould results in a positive replica, and vice versa. The bioimprints on PDMS master moulds and on CH were then compared using optical microscopy and AFM imaging.

### 2.5. AFM Imaging

A Digital Instruments (DI) 3100 Nanoscope III AFM from Vecco Instruments Inc. was used for imaging fixed cells and replica imprint samples in tapping mode, which allows a lateral resolution down to few nanometers. The X, Y and Z axial limits were 100 × 100 µm and 6 µm, respectively with a resolution 256 × 256 pixels. All images were taken at multiple sample positions and presented in an amber colour contrast scheme and were then processed using Gwyddion software.

### 2.6. Conductivity Measurement

The conductivity of crosslinked and non-crosslinked CH films was measured using four-point probes method employing a van der Pauw Hall effect measurement system under a magnetic field of 0.51 T. Film thickness, *t* was measured with a Surface DEKTAK 150 Profilometer (Veeco, New York, NY, USA). Resistance measurement was carried out three times for each sample. The resistivity *ρ* and conductivity *σ* are calculated as follows:Resistivity, *ρ* = *Rtw*/*l*
(1)
in which
Conductivity, *σ* = 1/*ρ*(2)
where *R* = resistance (Ω), *ρ* = resistivity of the material (Ω. cm), *Ɩ* = length of the film (cm), *w* = width of the film (cm), and *t* = thickness of the film (cm).

### 2.7. Swelling and Degradation Behaviours

Hydrophilicity and water uptake ratio are crucial properties in the consideration of biomaterials for cell culture studies and for tissue engineering applications. These properties are important for cell growth and adhesion as well as for absorption of body fluids and transfer of cell nutrients and metabolite functions. Moreover, biodegradable films with lower water uptake ratio show improved mechanical stability [63].

In this study, hydrated conductive films were immersed in purified water at room temperature until saturation and until no more swelling was observed (equilibrium). The swelling behaviour was measured by visual inspection and swelling analysis. The analysis was performed only on the 4% non-/crosslinked plasticized PEDOT:PSS in gelatin solution due to its excellent replication fidelity. The degradation of five concentrations of crosslinked CH films were quantified with respect to weight loss. This measurement was used in order to assess the CH substrates suitability for clinical applications as a tissue scaffold. The dry films with approximate weight of 2 g (*W_d_*) each were incubated at 37 °C in culture medium for 15 days. At each selected time point, films were removed from culture media, dried and measured the weight loss (*W_t_*).

The net weight loss, Ws percentage % was calculated using the following equation:Ws = ((*W_d_ − W_t_*)/*W_d_*) × 100(3)
where, *W_d_* and *W_t_* are the weights of the film at dry state and at each interval time, respectively.

### 2.8. Contact Angle

Water contact angle of the obtained substrates was carried out with telescope goniometer-CAMP 2008 KSV system. Four different concentrations of CH and control substrates (microscope glass, PDMS) were placed on the stage. On each substrate, a 5 µL droplet of purified water was dispensed and software automatically measured contact angle. Five measurements were taken from each sample and the optical images were analysed using Image J software. All readings were averaged to provide an averaged contact angle. One-way ANOVA analysis was performed to evaluate significance between multiple substrates.

### 2.9. Cell Adhesion

Mouse myoblast, C2C12 cells line were used to evaluate cell adhesion and viability when cultured on CH substrate. A cured 4% CH film was sandwiched between a glass substrate and a PDMS chamber assist by oxygen plasma at 100 W for 30 s to promote adhesion. The circular chambers were punched into a cured PDMS sheet using a cork borer with diameter of 15 mm. Three different substrates, microscope glass slide, PDMS and gold were chosen as control substrates and sealed with a PDMS chamber. All substrates were sterilised under UV irradation in a laminar flux cabinet for 60 min. Prior to cells seeding, the CH substrate was incubated in cell medium for 4 h and washed with PBS solution followed by immersion in culture medium to naturalise leaching of the CH film. About 5 × 10^4^ cells/mL of C2C12 cells were seeded on all substrates and incubated for 48 h. Cell viability was photographed after 24 h and 48 h of incubation.

### 2.10. Mechanical Properties

Tensile strength of four concentrations of CH substrates (2%, 3%, 4% & 6%) were measured employing a 858 Material Testing System. The tensile testing machine with pneumatic pressure controlled grips at a crosshead speed of 0.01 mm/min at room temperature were used for this purpose. A PDMS substrate was used as a control sample. Five samples for each substrates were measured and the significant values were analysed using one way ANOVA. 2.11 Statistical Analysis.

The differences in the means of the measured variables was compared with a one-way ANOVA analysis of variance where *p*-values smaller than 0.05 were considered statistically significant.

## 3. Results

This section presents the changes in conductivity, film thickness, replication fidelity, swelling behaviour, biodegradability analyses, wettability, biocompatibility and mechanical properties of bioimprinted CH as a result of adding gelatin, glycerol and mTg, as the cross-linker, to PEDOT:PSS. At least five samples were tested in each experiment to ensure reproducibility. Error bars are indicated in all histograms used in this section to show statistical variations.

### 3.1. Effect of Glycerol on Electrical Properties and Replication of Bioimprints on Plasticized PEDOT:PSS Thin Films

Using the four point probe technique and AFM imaging, the conductivity, thickness and surface roughness of plasticized PEDOT:PSS films were measured. The corresponding data are presented in Table 1. It was found that by increasing the glycerol content in the PEDOT:PSS thin films, the electrical conductivity and the thickness were also increased from 1.28 S/cm to 443 S/cm and from 90 nm to 125 nm, respectively.

To further understand the reason behind the enhancement in the electrical conductivity owing to the addition of glycerol, surface roughness of the samples was measured via AFM analysis. The root-mean-square roughness (R*_rms_*) of a surface is the most commonly evaluated parameter representing the surface roughness, measured in nm. The presence of glycerol had a noticeable effect on the surface roughness values. This is similar to Xiong and Liu’s work where they described the addition of dopant stimulating the PEDOT grains, which resulted in increasing the surface roughness [64]. These results indicated that the space between PEDOT and PSS chains was increased due to the increase in the molecular weight of the solvent. Hence, this led to an increase in the conductive polymer grain size and a reduction in surface roughness [64].

### 3.2. Analysis of Cellular Features Replica Made on Plasticized PEDOT:PSS (CP)

To investigate the feasibility of replicating cell-like features onto plasticized PEDOT:PSS thin films using PASE technique, three concentrations of glycerol (1%, 2% and 3% *w*/*w*) were used. From the AFM images of the bioimprinted plasticized PEDOT:PSS thin films, it was found that the cellular features were only successfully transferred in 4% glycerol.

Figure 3 shows optical microscopy images of the bioimprints replicated on CP films with two different magnifications, 10× and 20× for 2%, 3% and 4% of plasticized PEDOT:PSS concentrations in a gelatin solution. Figure 3a shows a low resolution of cell-like features for 2% concentration, where features are visible but very poorly replicated. There is also a formation of black dots on the PEDOT:PSS film, which are some debris from the PEDOT:PSS formation on the bioimprinted cellular features. However, for the 3% PEDOT:PSS, shown in Figure 3b, there is an improved replication fidelity. The optical image of bioimprints on 3 and 4% CP films, shown in Figure 3b,c, demonstrates that as the concentration of glycerol increased, the resolution of the features was improved. This results were, however, not ideal, as the film thickness was really low due to low viscosity of spin coated liquid plasticized PEDOT:PSS. In order to overcome this issue, gelatin was mixed with PEDOT:PSS to create a conductive hydrogel with increased film thickness. The results of replicating bioimprints onto CH will be explained in the following section.

### 3.3. Analysis and Enhancement of Cellular Features Replica on Coductive Hydrogel (CH)

A 4% concentration of glycerol and PEDOT:PSS in gelatin was selected as CH material for bioimprinting process. Both positive and negative bioimprints were prepared and used to quantify the cell replica morphology and they were compared to the original cell features. Figure 4 shows the optical images of the original cell features fixed on glass (a) and the 4% conductive bioimprint (b) for the same cells. These images reveal that the patterns are transferred but with low fidelity, resulting in a lack of some of the cellular features at micro- and nano-scale as compared with the original features.

The 2D profile of the AFM trace, as shown in Figure 5, illustrates the replication of the cell’s details in CH, where the height of the features is almost 70% lower than the height of the original. This loss was due to PDMS being hydrophobic and the hydrogel being water based, which resulted in loss of resolution in conductive bioimprint.

To improve the replication resolution of the conductive bioimprint, the following steps were taken. Firstly, the bioimprinted PDMS films were treated with oxygen plasma for 3 min at 30 W and then immersed in a 22.2% *w*/*v* solution of Polyvinylpyrrolidone (PVP) in deionized water for 1 min. PDMS moulds were finally washed thoroughly with deionized water and dried. The PVP treated PDMS substrate was liquid cast with CH mixture and cured on a hot plate at 50 °C for 1 h. The PVP is water-based hydrophilic and biocompatible polymer with good chemical stability. Here, PVP was used to help in turning the PDMS surface hydrophilic for a relatively long period of time. Therefore, PVP and additional curing steps helped hydrogel to better fill the patterned details and penetrate into the micro-and nanoscale features. The conductive bioimprint was left in the incubator at temperature of 37 °C overnight, then left at 4 ℃ for 48 h to further induce polymerization.

Figure 6a–c show the optical images of fixed cells, their negative bioimprints on PDMS and positive bioimprints on 4% CH after following the improved process described above. The red circles on the optical micrographs of the same cell indicate that the replication fidelity of the conductive bioimprint increased significantly when PDMS was PVP treated. AFM images were taken of the same cell to examine the resolution of the conductive bioimprint replica after treating PDMS with PVP. 

Figure 7a–e show the AFM images of the same cell fixed on glass, its negative and positive replication on PDMS, and its negative and positive imprint on CH. These images clearly show the high resolution of the cell imprint on CH. To prove this further, profiles of the cross section of the cell, fixed on glass, on PDMS and on CH, have been compared in Figure 8 and Figure 9.

The data demonstrate the successful replication of the cellular features with an average replication fidelity of more than 90% as compared to the original features. The conductive bioimprint fabrication process is able to replicate the cellular features without any deformation and artefacts, while maintaining the cell morphologies in micro- and nano-sized details.

### 3.4. Effect of the Crosslinker on the Conductivity of the Conductive Hydrogel Film

Figure 10 shows the effect of adding mTg to the CH mixture on the electrical conductivity for different concentrations of the CH films. It was found that the conductivities of non-crosslinked films (red graph) have extremely low conductivities ranging from 0.8 × 10^−6^ to 7.9 × 10^−6^ S/cm. However, the conductivities of the conductive films rose by six orders of magnitude after the crosslinking process (blue graph), to values ranging from 1.2 to 2.1 S/cm. Moreover, with *p*-value smaller than 0.00001 shows that the addition of mTg crosslinker significantly enhanced the conductivity of crosslinked films. Additionally, the more conductive the hydrogel is, the more sensitive the film is to changes in cell growth as an increase cell numbers is translated to an increase in impedance. A conductivity of 1 S/cm is suitable for monitoring cells growth as a function of conductivity [63].

### 3.5. Effect of Crosslinker on the Swelling Capacity of Conductive Hydrogel Film

In general, the unique polymer network structure of hydrogels t exhibit significantly different swelling functionalities. Swelling behaviour is an intrinsic property of the CH films when used as a cell culture platform. The swelling problem of the conductive hydrogel network could lead to loss of the final composite material and change the electrical conductivity due to the percolation circumstance. By improving the mechanical properties of CH films via crosslinking, the material possesses tighter internal entanglement structure which creates a denser interconnection network in the polymer matrix. Thus, a synthesized conductive hydrogel composite could maintain its electrical properties and higher its water resistance with enhanced the mechanical performance by using the mTg crosslinker. To examine the water absorption capacity, the non-crosslinked and crosslinked CH film were immersed in deionized water for certain lengths of time as shown in Figure 11. The 4% non-crosslinked CH film was swollen to three times its original size after two hours in water and was totally dissolved after less than four hours.

As can be seen in Figure 11b, the crosslinked CH films demonstrated that lower water absorption swelling behaviour and completely dissolved at 14 days after being soaked in deionized water.

The percentage swelling of the 4% CH (non-crosslinked and crosslinked) films is shown in Figure 12. It can be seen that the CH prepared without the crosslinker swelled to reach a weight 150% higher than its original weight after two hours and then started to dissolve due to the weakening of its chemical bonding. The crosslinked film, however, showed lower water absorption, much less swelling and reached equilibrium saturation condition after 3 h in deionized water, at only 50% higher than its original weight. Addition of mTg crosslinker to CH, resulted in formation of a denser structure and limited diffusion process which then led to a decrease in swelling rate.

In summary, the optimum conductive polymer material suitable for conductive bioimprint applications is the 4% *w*/*w* of PEDOT:PSS and 4% *w*/*w* of glycerol into 12.5% hydrogel mixture which gave the best combined properties of high replication fidelity, high conductivity, and less swelling behaviour.

### 3.6. Effect of Crosslinking on the Biodegradation Properties of the Conductive Hydrogel Film

The biodegradation behaviour of the CH films was examined in vitro. Films were immersed and incubated in a culture medium at 37 °C and monitored over time. Figure 13 shows the percentage weight loss of the non-/crosslinked CHs films measured over a period of 15 days. The degradation results revealed that the presence of PEDOT:PSS and glycerol significantly slower the degradation processes of the crosslinked CH films. Hydrogel films prepared with 0% concentration of PEDOT:PSS showed a higher percentage of degradation and completely degraded in 11 days compared to other films. At 2% and 3% of CH, films lost about 87.8% to 93% of their weight at the end of the 15-day period, whereas a continued slow weight loss was observed for the 4% and 6% concentrations of the CH films.

### 3.7. Wettability Properties

Figure 14 shows the water contact angles measured for seven different substrate surfaces; glass, PDMS, gold and CH substrates with different concentration. The average water contact angle of the PDMS substrate was about 113.8 ± 3.0°. The ANOVA (*p* < 0.0001) indicated a notable hydrophobic properties compared to the other substrates. In contrast, the glass slide and gold substrate showed a good water wettability properties with contact angles of 50.2 ± 1.7° and 62.2 ± 4.8°, respectively. Similarly, the 2%, 3%, 4% and 6% concentrations of conductive hydrogel films exhibited hydrophilic properties with contact angles of 62.5 ± 1.3°, 60.6 ± 1.7°, 66.7 ± 1.2° and 56.2 ± 3.0°, respectively. The analysis shows that the *p* > 0.05 of the water contact angles of CH substrates are not significantly different from the control glass and gold substrates.

### 3.8. Biocompatabilty Properties of Conductive Hydrogel Film

After incubation for 24 h and 48 h in cell culture medium, the substrate’s surface was imaged using an optical microscope. The images of cells grown on different flat substrates: glass slide, PDMS, gold and 4% concentration conductive hydrogel are shown in Figure 15. At 24 h, cells grown on the 4% CH showed an extension of the membrane protrusions and were well spread, whereas very few cells remained adhered on the PDMS substrate surface after 24 h. After 48 h, the C2C12 cells grown on the conductive hydrogel were well-spread and reached confluence. Although PDMS is a biocompatible substrate, cells did not reach confluency due to their hydrophobic properties. Hence, after 48 h, the C2C12 cells begun to attach and spread on the PDMS substrate. At 24 h and 48 h, cells cultured on the glass slide revealed likewise morphologies to those on the gold substrate. The biocompatibility analyses confirmed that the CH substrates used are suitable for culturing living cells. The high rates of cell viability after contact with the CH’s platform confirm that CH substrates are not biologically harmful.

### 3.9. Tensile Test

Figure 16 presents the tensile test of the conductive hydrogel films and PDMS as a control film. As can be seen, the average elastic modulus of conductive films was within the range of 1.08–4.55 MPa and increased by increasing the percentage of PEDOT:PSS and glycerol. One-way ANOVA analysis shows that the *p*-values for 3%, 4% and 6% of CH films are larger than 0.05 considered not significantly different with PDMS substrate, whereas 2% concentration of CH film revealed an elastic modulus of 4.55 MPa compared to the control PDMS film of 1.58 MPa. The mechanical properties of 2% CH film showed less stretchable properties among other films with *p*-values smaller than 0.005 were considered statistically significant. The presence of PEDOT:PSS and glycerol in the hydrogel matrix has significantly improved their mechanical properties. This is in agreement with the biodegradation test discussed in Section 3.6, which illustrate that films containing PEDOT:PSS have a more stable structure.

## 4. Discussion

The aim of this study was to develop a device for monitoring cell behaviour as a function of their impedance based on bioimprint technology and CP. This is the first report on a CH cell-culture platform replicated with cellular morphology. The bioimprints help guide growth and differentiation of the culturing cells, while the conductive substrate allows for measuring and monitoring cell impedance which is indicative of cell viability and spread. This can be modified and applied to monitor the success and progress of medical implants in real-time [65]. Currently, medical implants are judged by patients’ comfort or discomfort. This might lead to complications and infections if problems are not realised early enough.

Upon addition of glycerol to PEDOT:PSS, the electrical properties of the plasticized PEDOT:PSS thin films improved and the conductivity increased by two orders of magnitude. One possible explanation is that the existence of glycerol breaks the ionic bond in the PEDOT:PSS chains, so the PEDOT particles evinced more aggregation and created a sturdy bond between the counter ions in the chains [42,66]. Huang et al. [28] reported that the electrical conductivity of doped CP films increases by two- to threefold due to the identical distribution of PEDOT molecules in the PEDOT:PSS films and produced well defined pathways for charge transport perpendicular to the film.

However, the resolution of the replicas was very low and the replicas lost most of the micro- and nano-sized details. The poor transfer was possibly due to the very low viscosity and thin plasticized PEDOT:PSS films, as the thickness ranged from 90 nm to 125 nm. This made the imprinting process harder when the PDMS mould pattern is pressed against the target film. It is concluded that with the addition of glycerol, the electrical conductivity of the thin film was increased, but the resulting films were not suitable for transferring cell-like features. This is due to low viscosity and thickness problems, which caused large loss of the micron and nano sized cellular details. The importance of retaining the cell features and resolution is to keep the cell-like topography details on the culture platform to promote cell preferential attachment and guidance and facilitate conductivity change monitoring.

In order to improve the low viscosity and film thickness issues, an alternative method was developed in this work by introducing gelatin to achieve a higher replication fidelity and thicker films, as well as being biocompatible. The newly developed CH mixture was also crosslinked via addition of mTg to the solution prior to liquid casting on a PDMS master mould to form a freestanding conductive bioimprinted film. Crosslinking improved mechanical properties of the CH, such as swelling and degradation, while also significantly increasing the conductivity by six orders of magnitude.

This improvement in the conductivity was due to a denser interconnections network in the polymer matrix. This is in agreement with Zhao et al. [27], where they reported that the conductivity of another electrically conductive hemicellulose hydrogel (ECHH) film has increased by three orders of magnitude after it was crosslinked with 10–40% Aniline Pentamer (AP) crosslinker. They found that there was a strong relationship between the total concentration of the CP and the proportion of the crosslinker in the hydrogel. Increasing the AP concentration resulted in a higher whole concentration in the hydrogel, which enhanced the conductivity of the ECHH film.

Swelling behaviour of the CH films was an important property to be tested before using the hydrogels as cell culture substrates. Swelling is an indicative measure of water absorption and leads to degradation of the structure. This is especially relevant considering that the PEDOT:PSS is water soluble and would dissolve in typical cell culture media. Typically, hydrogel can exhibit different in their swelling behaviour, network structure, a degradation and mechanical strength in response to different synthetisation method.

By crosslinking the hydrogel with a microbial transglutaminase (mTg), the swelling behaviour of the film, as measured by water absorption and weight changes, was improved and the resultant crosslinked hydrogel exhibited water resistance suitable for cell culture platforms.

The significantly different swelling behaviours of the modified hydrogel illustrate that the cross-linked CH film possesses tighter internal entanglement structure and a homogeneous distribution of crosslinking centres as compared to the non-crosslinked CH film [20,21].

## 5. Conclusions

In this work, we developed a conductive bioimprint technique, a novel 3D replication of cellular morphology onto a CH matrix based on PEDOT:PSS. A soft lithography technique was employed in the process to form a conductive bioimprint that possesses desirable properties for functional cell culture platforms such as good conductivity, biocompatibility, stretchability, high replication resolution, and water resistance [63]. The conductive bioimprint offers surface topography of cell-like features which can be used as an implant coating to improve the interface between the implant and body tissues. We have found in our previous studies [2,3,4,9,13,14,15] that cells prefer to grow on the imprinted features compared to flat surfaces. In addition, the high conductivity allows monitoring the electrical signal of living cells which give an indication of cell attachment and growth on medical implants in real-time. This helps with observing medical implant progress and success, and might prevent complications due to infections at early stages, which is not currently available in medical implants. The degradation properties of the coating, on the other hand, are an added advantage for the implant, as it is a sacrificial layer that will disintegrate after guiding and promoting the cell growth.

The addition of glycerol with different concentrations significantly increased electrical conductivity and surface morphology of the PEDOT:PSS thin films. Moreover, bioimprints on CP films without gelatin fabricated using PASE method exhibited noticeable loss of resolution due to low thickness of the films. When gelatin was incorporated into the PEDOT:PSS along with glycerol, a much improved replication resolution was obtained. The AFM imaging employed in this study confirmed that the replication of the cellular footprint of the C2C12 mouse myoblast captured more than 90% of the original cell details down to the micro- and nanometre. By adding the mTg crosslinker, the electrical conductivity was further enhanced, by six orders of magnitude, and the water resistance of the film also improved. The developed CH material is based on PEDOT:PSS and is suitable for producing high resolution replicas for cell interaction studies. This highly conductive substrate with patterned surface is suitable for biosensing and monitoring of cell behaviour as a function of conductivity in addition to being biocompatible and biodegradable. The conductive hydrogel has the potential to be used in monitoring medical implants’ progress in real time.

## Figures and Tables

**Figure 1 bioengineering-08-00204-f001:**
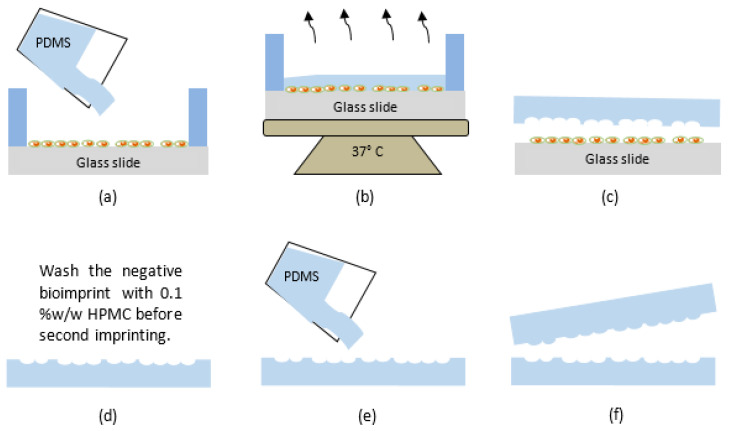
Schematic diagram of negative and positive PDMS bioimprint fabrication process: (**a**) liquid PDMS is dispensed on fixed cells, (**b**) the polymer is cured at 37 °C for 12 h, (**c**) negative PDMS mould is carefully peeled off from fixed cells and post-baked at 80 °C for two hours to complete polymerization, (**d**) negative PDMS mould is washed with 0.1% *w*/*w* HPMC for 10 min, (**e**) liquid PDMS is poured on the negative mould, then the curing process follows as in in step (**b**), and (**f**) the positive bioimprint PDMS is peeled off from negative PDMS bioimprint mould and hard baked at 80 °C for two hours.

**Figure 2 bioengineering-08-00204-f002:**
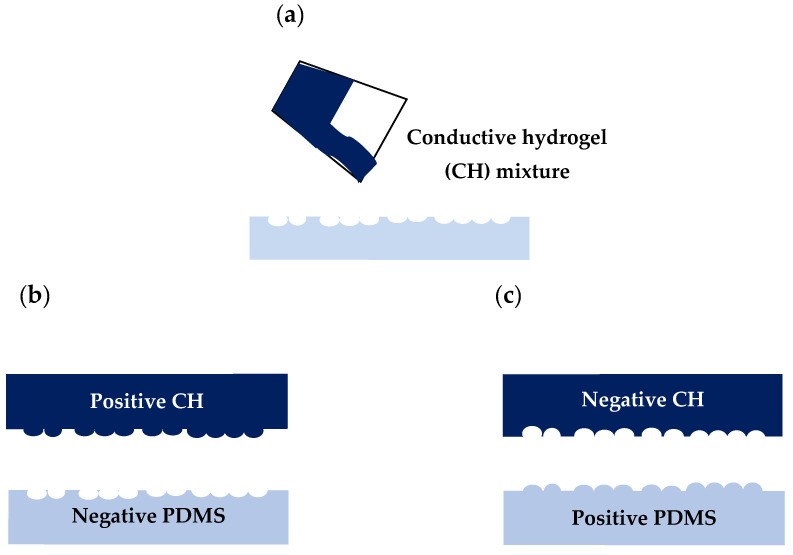
Schematic of conductive hydrogel with positive and negative bioimprints: (**a**) CH mixture is liquid cast on PDMS moulds with negative and positive bioimprints and left in the incubator at 37 °C overnight, then further cured at 4 °C for 48 h, (**b**) negative and positive CH moulds are peeled off the PDMS mould, (**c**), positive CH and negative PDMS replicas produced by soft lithography process.

**Figure 3 bioengineering-08-00204-f003:**
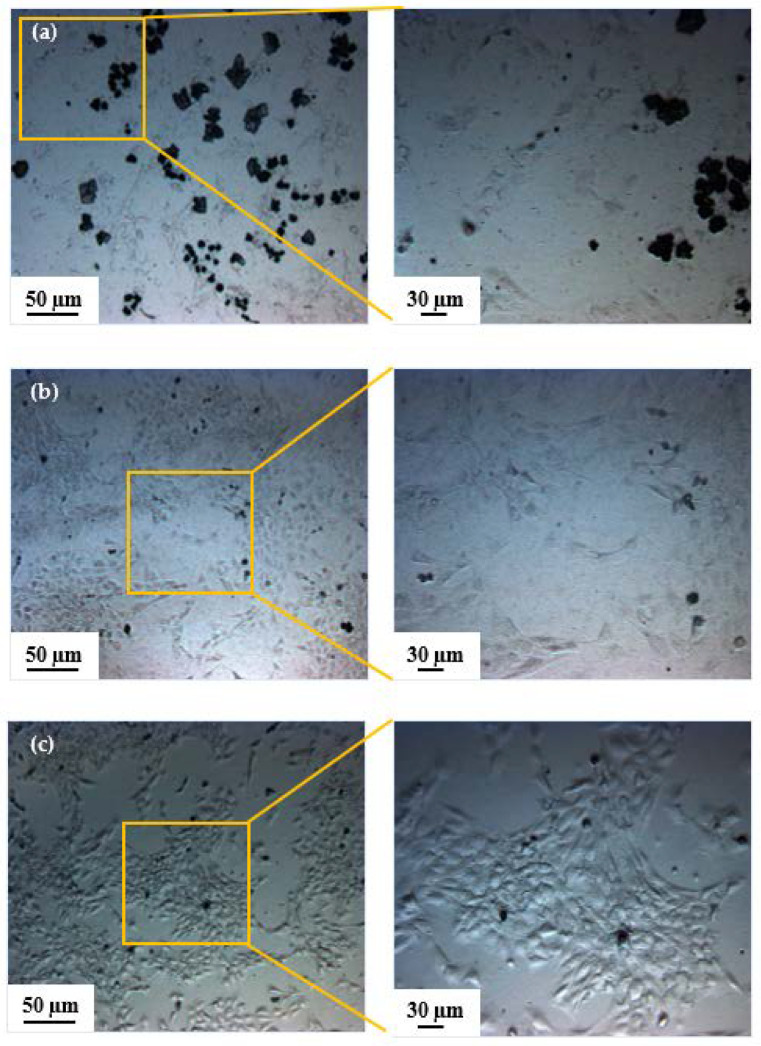
Optical images of replicated cells onto conductive polymers with different concentrations of glycerol: (**a**) 2%, (**b**) 3% and (**c**) 4%. By increasing the percentage of glycerol in PEDOT:PSS the replication resolution was improved. The black spots in (**a**) are some debris from the PEDOT:PSS formation on the bioimprinted cellular features.

**Figure 4 bioengineering-08-00204-f004:**
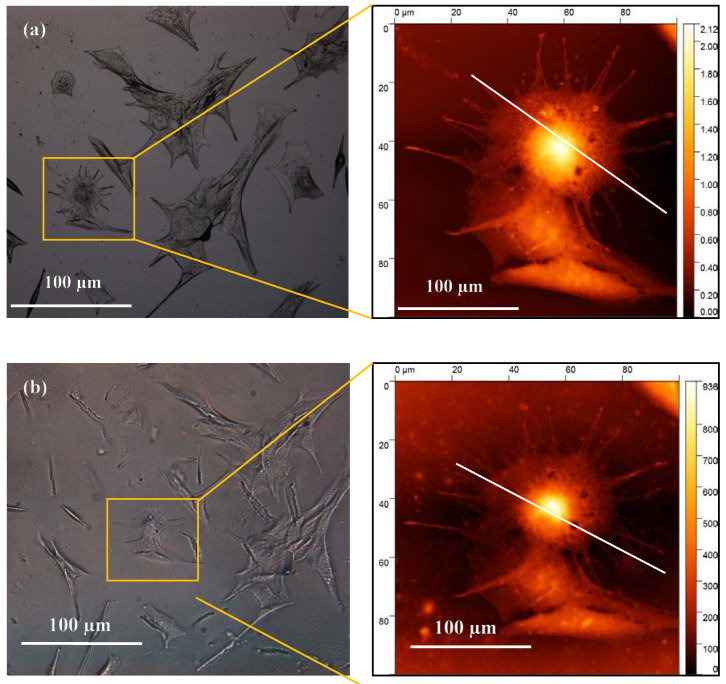
On the left, optical micrographs of (**a**) fixed C2C12 cells on glass and (**b**) positive replication in 4% conductive hydrogel. On the right are the AFM images of a single cell on the same substrates. The white lines across the cell indicate the profile of the cross-section of the cell at each step, which are compared in the next figure.

**Figure 5 bioengineering-08-00204-f005:**
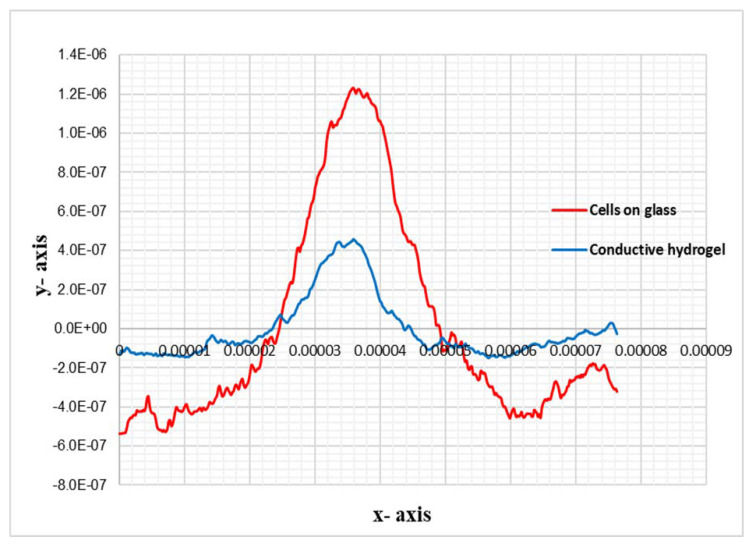
Topographic profiles of cross section of the cell shown in Figure 4, across the white line on fixed cell and its positive imprint on conductive hydrogel, without PVP treatment of PDMS, y axis scale units is in meter.

**Figure 6 bioengineering-08-00204-f006:**
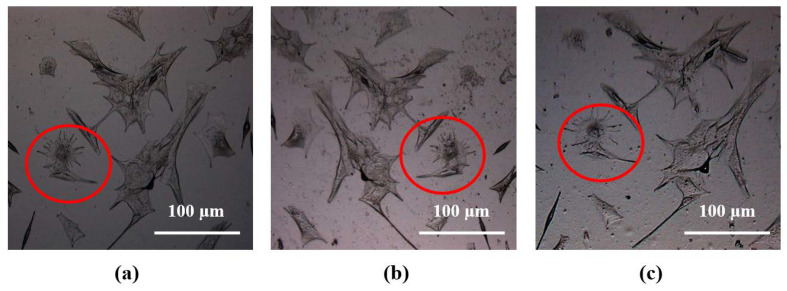
Optical images of (**a**) fixed C2C12 cells on glass, (**b**) negative replica in PDMS, and (**c**) positive replica in conductive hydrogel after plasma and PVP treatment of PDMS. The red circles highlight a unique cell that was used in all the AFM images represented in the next figure.

**Figure 7 bioengineering-08-00204-f007:**
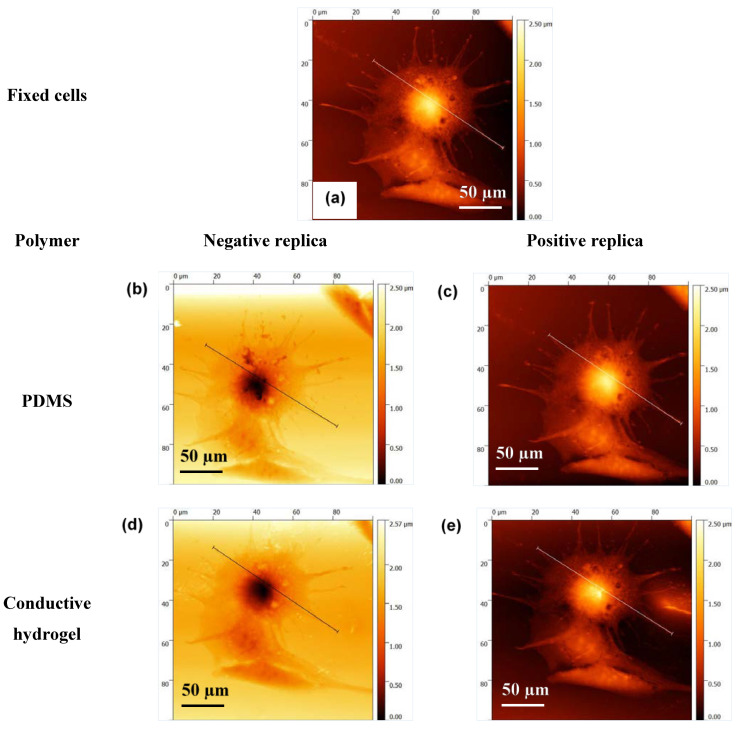
AFM images of a single C2C12 cell (**a**) fixed on the glass, (**b**,**c**) its negative and positive imprints on PDMS and (**d**,**e**) its negative and positive imprint on conductive hydrogel after plasma and PVP treatment of PDMS. The lines across the cell indicate the profiles of the cross-section of the cell at each step, which are compared in the next figure.

**Figure 8 bioengineering-08-00204-f008:**
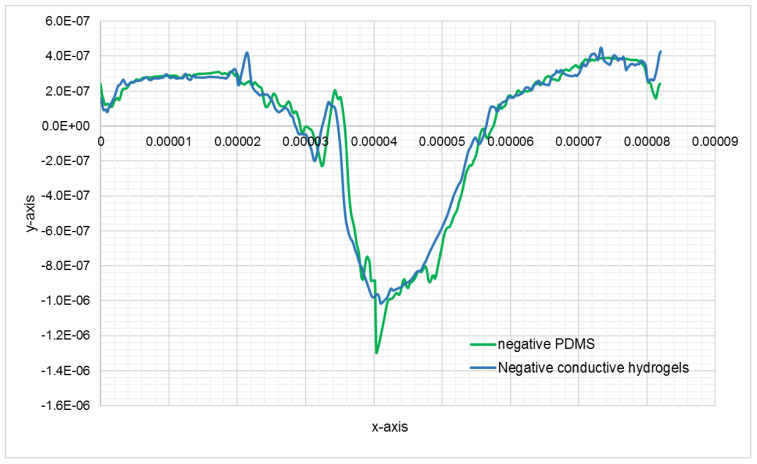
Comparison of the topographic profiles of the cross section of the cell shown in Figure 8, on negative PDMS bioimprint and negative conductive hydrogel bioimprint, y scale units is in meter.

**Figure 9 bioengineering-08-00204-f009:**
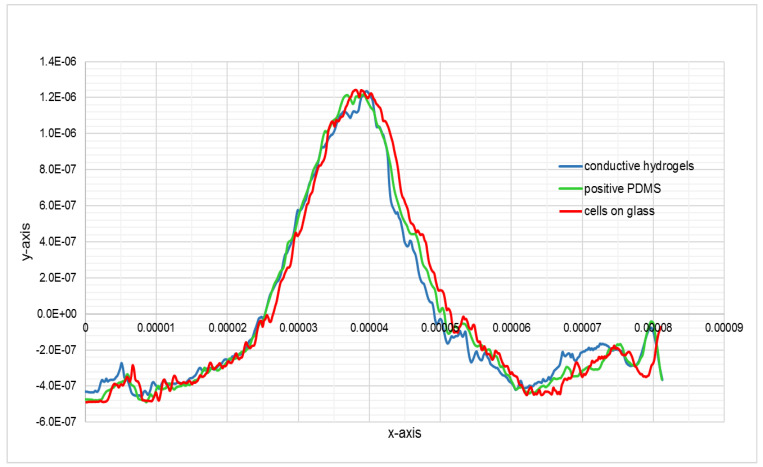
The topographic profiles of the cross-section of the cell shown in Figure 8, extracted from AFM data of fixed cell, positive PDMS bioimprint and positive conductive hydrogel bioimprint, the units in the y scale is in meter.

**Figure 10 bioengineering-08-00204-f010:**
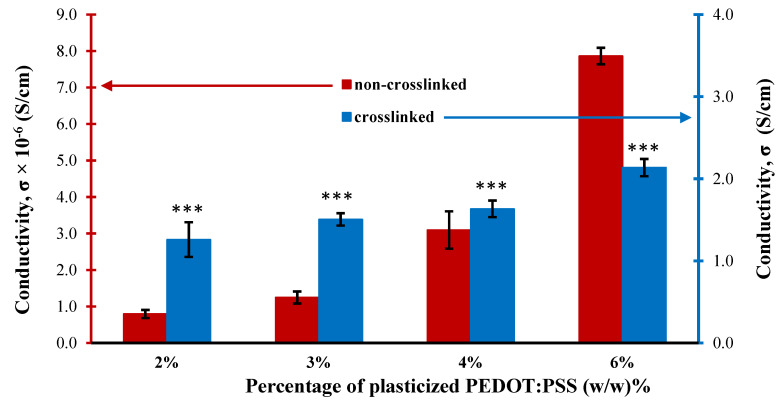
Electrical conductivity of non-crosslinked and crosslinked conductive hydrogel films. Denotes *** *p* < 0.00001.

**Figure 11 bioengineering-08-00204-f011:**
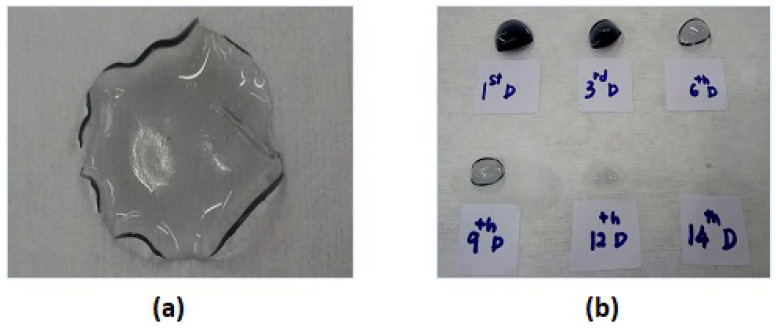
Swelling behaviour of (**a**) non-crosslinked conductive hydrogel film and (**b**) crosslinked conductive film.

**Figure 12 bioengineering-08-00204-f012:**
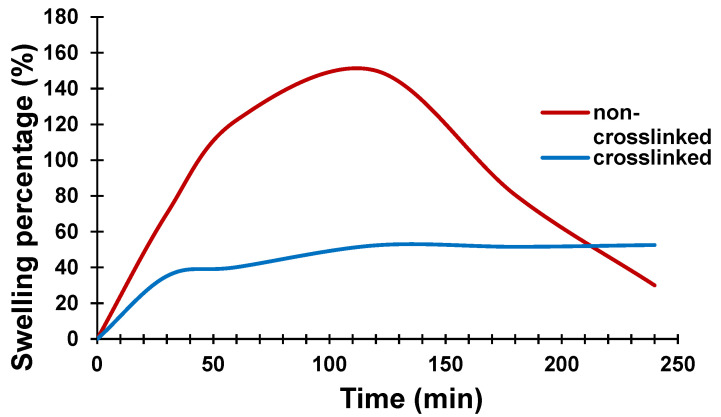
Swelling percentage of the non-crosslinked and a crosslinked conductive film.

**Figure 13 bioengineering-08-00204-f013:**
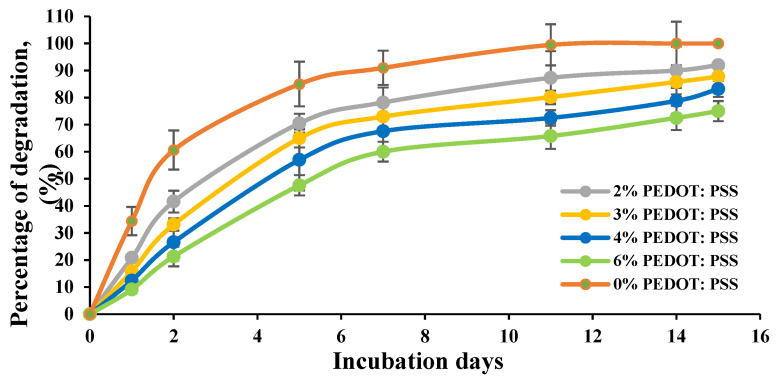
Degradation as measured by the weight loss of the CH films soaked in cell culture medium. The existence of PEDOT:PSS improved the CH degradation behaviour significantly.

**Figure 14 bioengineering-08-00204-f014:**
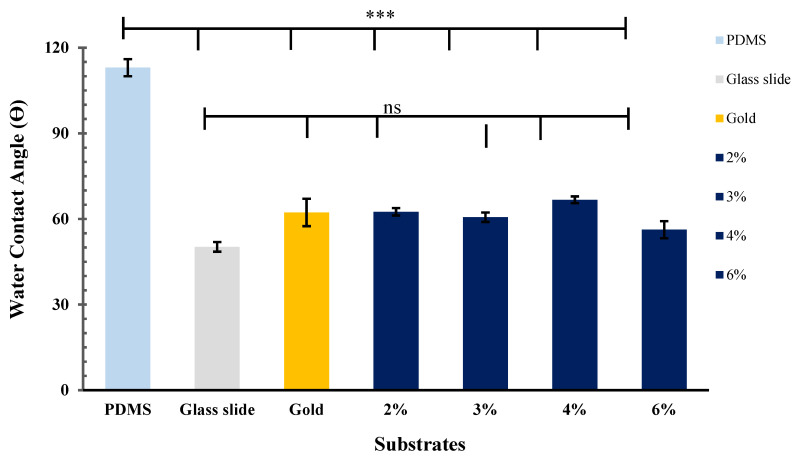
The graph indicates that the CH surfaces have suitable wettability with an average contact angle ranging between 56.2 ± 3.0° to 66.7 ± 1.2°. Denotes *** *p* < 0.0001; ns = not significant with *p* > 0.05; *n* = 5.

**Figure 15 bioengineering-08-00204-f015:**
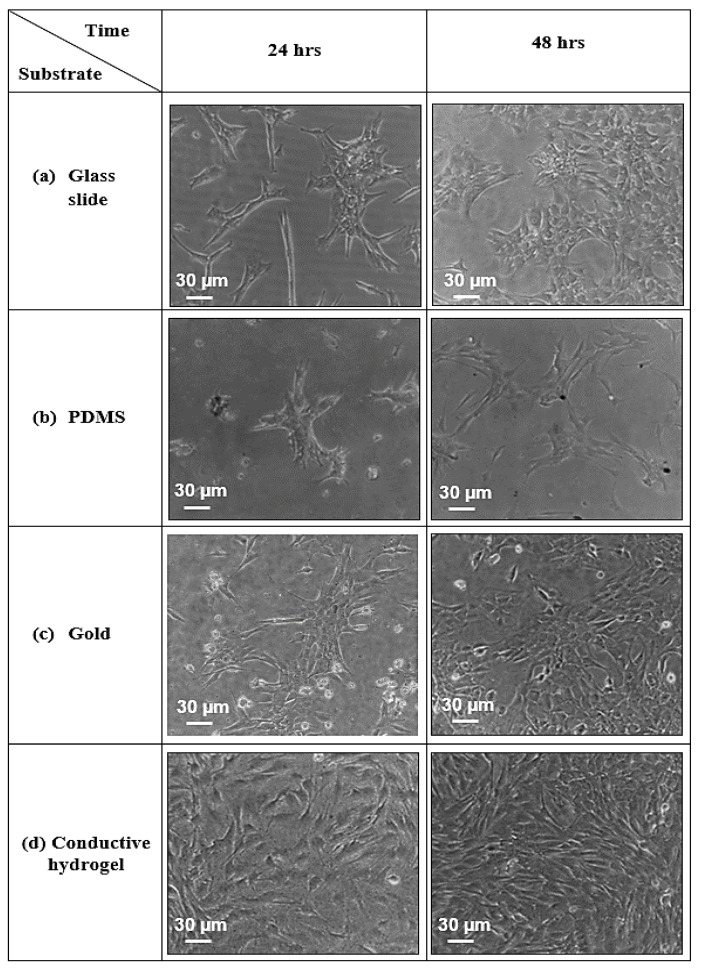
The growth of C2C12 cells for 24 and 48 h on different substrates. The image shows cells growing on (**a**) glass slide (control), (**b**) PDMS (control), (**c**) gold (control) and (**d**) 4% CH.

**Figure 16 bioengineering-08-00204-f016:**
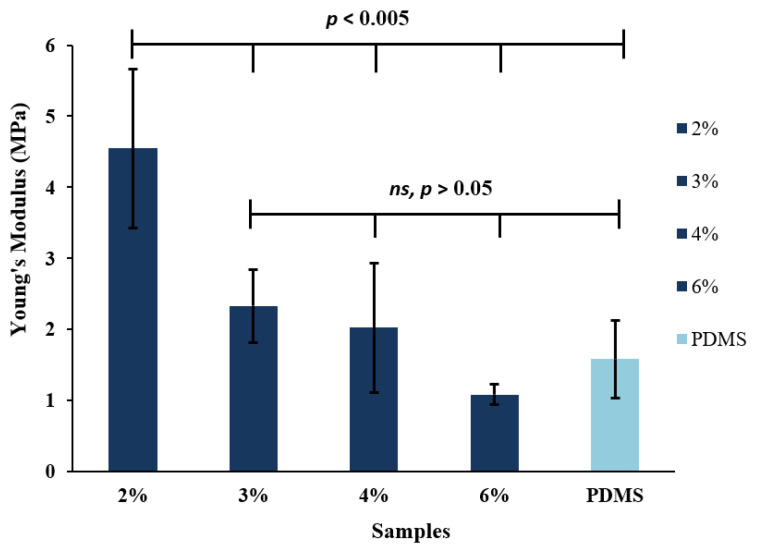
Tensile test of conductive hydrogel films and PDMS film. The data show a significant increase in Young’s Modulus as the percentage of PEDOT:PSS and glycerol decreased in the hydrogel matrix. Denotes *n* = 5 samples for each film.

**Table 1 bioengineering-08-00204-t001:** The thickness (*t*), conductivity (σ), resistivity (ρ) and surface roughness (R_rms_) of the films. The conductivity, σ of plasticized PEDOT:PSS thin films significantly increased with the addition of glycerol, suggesting that the presence of glycerol helped in rearranging the PEDOT:PSS morphology and improved interconnections between the PEDOT chains.

Percentage of Glycerol in PEDOT:PSS	Thickness (nm)	Conductivity σ (S/cm)	Resistivity ρ (Ω.cm)	R*_rms,_* nm
0	90 ± 1.7	1.28	0.782	1.878
0.5	105 ± 1.0	3.05 × 10^2^	3.28 × 10^−3^	2.219
1	114 ± 1.3	3.24 × 10^2^	3.09 × 10^−3^	2.617
2	119 ± 1.3	3.91 × 10^2^	2.56 × 10^−3^	3.829
3	122 ± 1.2	4.16 × 10^2^	2.40 × 10^−3^	3.865
4	125 ± 1.3	4.43 × 10^2^	2.26 × 10^−3^	4.568

## Data Availability

Data available in a publicly accessible repository. The data presented in this study are openly available in UC research repository “Abd Wahid, N.A. Conductive Bioimprint for Regenerative Medicine: Synthesis and Characterisation. PhD thesis, University of Canterbury, Christchurch, New Zealand, 13 May 2019”. http://dx.doi.org/10.26021/3027.

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
