# Peer review of "Conductive Bioimprint Using Soft Lithography Technique Based on PEDOT:PSS for Biosensing"

_bioengineering, 2021, doi:10.3390/bioengineering8120204_

Round 1
Reviewer 1 Report
The manuscript deals with the preparation and characterisation of conductive cell surface replicas based on soft lithography applied to PEDOT-PSS blends with glycerol and gelatine, with and without biochemical cross linking. The replicas will in turn be used in applications such as cell proliferation monitoring through e.g. impedance measurements. This is an interesting topic in line with the journal readership, to add monitoring capabilities to implants, for example to monitor potential fouling processes after surgery. The conclusions are sound and well supported by experimental results. The manuscript can be accepted after minor revisions and answering the following comments:
In one case hemicellulose is used to assist replication of the morphological surface features, and in a second case, PVP is applied. Could you please discuss/comment further why the type of treatment was changed? Also, what is the effect of these additives on future cell adhesion? Are there studies demonstrating improvement or decrease in cell adhesion to the resulting surfaces? PLL or phospholipids could for example be used instead of PVP to get a more natural interface layer.
P2, line 63, “structural bone replacement to electrically stimulate the neural 63
probes”, could you please explain the link between the two in more detail, perhaps two sentences/concepts were mistakingly merged or a term is incorrect?
L 486: (…) resulted in a higher hole concentration in the, which(…) there is something missing after “The”
The quality of microscope images is very low (pixelated in many cases) and should be improved in the final version. Please also consider applying similar graphical layout and improving the quality of figures (fonts, graph area, presence/absence of grids etc.) throughout the manuscript.
Author Response
- In one case hemicellulose is used to assist replication of the morphological surface features, and in a second case, PVP is applied. Could you please discuss/comment further why the type of treatment was changed? Also, what is the effect of these additives on future cell adhesion? Are there studies demonstrating improvement or decrease in cell adhesion to the resulting surfaces? PLL or phospholipids could for example be used instead of PVP to get a more natural interface layer.
The use of PVP in this work was only to functionalize the surface of PDMS making it hydrophilic for a longer period of time, such that the PEDOT:PSS-gelatin solution (which is water-based) would wet the surface of PDMS in more effective way, resulting in a significant improvement in the resolution of bioimprints on the final sample. PVP is not transferred onto the gelatin films, since any residual on the PDMS moulds is washed away before liquid casting gelatin solution, therefore PVP does not have any effect on cell adhesion or their behavior.
Hemicellulose was not used in this work and it was only mentioned as a comparison to crosslinking of the PEDOT:PSS hydrogel.
The biocompatibility analyses confirmed that all the materials used in the fabrication of our films are safe in contact with living cells. The high rates of cell viability after contact with the developed conductive hydrogel CH’s extracts indicate that they are biocompatible.
- P2, line 63, “structural bone replacement to electrically stimulate the neural probes”, could you please explain the link between the two in more detail, perhaps two sentences/concepts were mistakenly merged or a term is incorrect?
L 63: The manuscript has been changed to “Other studies on structural bone replacement, in order to electrically stimulate the neural probes, have led to the development of a wide range of prosthetic and medical implant devices.”
- L 486: (…) resulted in a higher hole concentration in the, which(…) there is something missing after “The”
L 486: the word “hydrogel” was missing and is added as follows: Increasing the AP concentration resulted in a higher hole concentration in the “hydrogel”, which enhanced the conductivity of the ECHH film.
- The quality of microscope images is very low (pixelated in many cases) and should be improved in the final version. Please also consider applying similar graphical layout and improving the quality of figures (fonts, graph area, presence/absence of grids etc.) throughout the manuscript.
Have addressed all the images raised by the reviewers. A new original and high resolution images are provided in separate files.
Reviewer 2 Report
This study developed a biocompatible conductive hydrogel with bioimprinted surface features. The authors presented the bioimprinted conductive hydrogel with high conductivity, biocompatibility and high replication fidelity and they considered it is suitable for cell culture applications. Since the bioimprinted conductive hydrogel is developed to investigate biological cells’response to their morphological footprint and study their growth, adhesion, cell-cell interactions and proliferation as a function of conductivity, It is considered that the data quality of this study need to improve.
- The authors need to show the mechanical properties and hydrophobicity of the hydrogels with different concentrations of PEDOT: PSS.
- Besides single cell image, the author also needs to show the cell viability of the conductive hydrogel
- For implant issue, the author didn’t show any proof of the biodegradable properties.
Author Response
Reviewer 2 Response is attached

Round 2
Reviewer 2 Report
English needs editing and figures need to combined.
Author Response
Reviewer 2 response is attached
